# Deep Learning-Based Acoustic Echo Cancellation for Surround Sound Systems

Guoteng Li [1,2], Chengshi Zheng [1,2], Yuxuan Ke [1,2,*] and Xiaodong Li [1,2]

1. Key Laboratory of Noise and Vibration Research, Institute of Acoustics, Chinese Academy of Sciences, Beijing 100190, China
2. University of Chinese Academy of Sciences, Beijing 100049, China
* Correspondence: keyuxuan@mail.ioa.ac.cn

**Abstract:** Surround sound systems that play back multi-channel audio signals through multiple loudspeakers can improve augmented reality, which has been widely used in many multimedia communication systems. It is common that a hand-free speech communication system suffers from the acoustic echo problem, and the echo needs to be canceled or suppressed completely. This paper proposes a deep learning-based acoustic echo cancellation (AEC) method to recover the desired near-end speech from the microphone signals in surround sound systems. The ambisonics technique was adopted to record the surround sound for reproduction. To achieve a better generalization capability against different loudspeaker layouts, the compressed complex spectra of the first-order ambisonic signals (B-format) were sent to the neural network as the input features directly instead of using the ambisonic decoded signals (D-format). Experimental results on both simulated and real acoustic environments showed the effectiveness of the proposed algorithm in surround AEC, and outperformed other competing methods in terms of the speech quality and the amount of echo reduction.

**Keywords:** acoustic echo cancellation; surround sound; ambisonics

## 1. Introduction

Surround sound systems offer the potential for immersive sound field reproduction [1], enhancing realism in virtual reality and multimedia communication systems [2], such as immersive teleconference and acoustic human–machine interfaces. In a closed-loop tele-conference system, the echo signal caused by the acoustic coupling between microphones and loudspeakers has a significant negative impact on hands-free speech communication systems. Conventional AEC methods often use adaptive filters to identify the acoustic echo paths, and the echo signal in each microphone is then estimated and subtracted from each microphone signal [3,4]. However, the adaptive filtering-based algorithms may suffer from the well-known non-unique solution problem [5] due to high cross-correlation between the loudspeaker signals, when there are two or more reproduction channels. Although many algorithms [6–9] have been proposed to decorrelate the loudspeaker signals so as to solve the non-unique solution problem, the reproduction quality and immersion of the far-end may be affected. Besides, as the number of channels increases, the computational complexity and convergence time will increase, and the control of the step size becomes much more sophisticated [10].

In recent years, deep learning-based methods have been employed in AEC and have achieved significant success. Compared with conventional AEC algorithms, deep learning-based AEC methods have the ability to recover the near-end signals from the microphone signals directly, and they do not need to identify the acoustic echo paths explicitly and also do not suffer from the non-unique solution problem. Lee et al. [11] proposed a deep neural network (DNN)-based residual echo suppression gain estimation, which

was then used to remove the nonlinear components after a linear acoustic echo canceller. Zhang et al. [12] formulated AEC as a supervised speech separation problem and proposed a recurrent neural network with bidirectional long short-term memory (BLSTM) to separate and suppress the far-end signal, hence removing the echo. However, due to the non-causality of BLSTM, the usage of this method may be restricted, especially in real-time applications. Cheng et al. [13] proposed a convolutional recurrent network (CRN) model to estimate the non-linear gain from the magnitude spectra of both the microphone and far-end signals for the stereo AEC, which was then multiplied by the spectrum of the microphone signal to estimate near-end speech. Besides, Peng et al. [14] described a three-stage AEC and suppression framework for the ICASSP 2021 AEC Challenge, where the partitioned block frequency domain least mean square (PBFDLMS) with a time alignment was firstly implemented to cancel the linear echo components, and two deep learning networks were then proposed to suppress the residual echo and the non-speech residual noise simultaneously. In addition, Zhang et al. [15] proposed a neural cascade architecture, including a CRN module and an LSTM module, which is used for joint acoustic echo and noise suppression to address both single-channel and multi-channel AEC problems. More recently, Cheng et al. [16] proposed a deep complex multi-frame filtering network for stereophonic AEC, where two deep learning-based modules were separately used for suppression of the linear and residual echo components.

Nowadays, with the widespread adoption of virtual reality and immersive teleconference, the surround sound-based real-time communication system will be popularized in the future trend. It is well known that the Ambisonic technique [17–19] is one basic and common surround sound recording and reproduction approach, where the Ambisonics encoder decomposes a sound field signal into spherical harmonics, i.e., B-format, and the Ambisonic decoder transforms the B-format signal into a multi-channel sound field signal, i.e., D-format, and they are then played back by multiple loudspeakers with a special layout, for example, the 5.1 channel surround sound layout. To the best of our knowledge, Ambisonics-based surround AEC methods have not been well studied yet. This paper proposes a gated convolutional recurrent network (GCRN) model to suppress the echo signal for the surround sound reproduction system. The input features of the GCRN model are the compressed complex spectra of the microphone signal and the B-format signals of the far-end instead of the D-format signals. This setting is mainly under the consideration that the actual loudspeakers layout in the near-end room may be different from the desired layout due to the obstacles or artificial errors, leading to the mismatch of the Ambisonic decoding matrix. The output of the GCRN model was the compressed complex spectrum of the near-end speech and the cost function was calculated between the estimated and real near-end speech signals with regard to their real and imaginary parts of the compressed complex spectra, respectively. The proposed algorithm was evaluated using the perceptual evaluation of speech quality (PESQ) [20] in double-talk scenarios and echo return loss enhancement (ERLE) in single-talk scenarios. The GCRN-based algorithm showed its effectiveness in both the simulated and real acoustic environments. In summary, this paper has two main contributions. On the one hand, the surround AEC is taken into consideration and the Ambisonics technique is adopted to record the surround sound for reproduction. On the other hand, the B-format signals instead of the D-format signals are used as the references to achieve better generalization against different non-standard loudspeaker layouts.

The rest of this paper is organized as follows. A brief introduction to the fundamentals of Ambisonics is described in Section 2. The surround AEC problem and symbols definition are formulated in Section 3. Section 4 presents the used network architecture and the compressed complex spectrum. Experimental settings and results as well as analysis are given in Section 5. In Section 6, we give our conclusions.

## 2. First-Order Ambisonics (B-Format)

Ambisonics is unique in being a total systems approach to reproducing or simulating the spatial sound in all its dimensions [17] and the Ambisonic encoder decomposes the sound field at a particular point of space on the orthogonal basis of spherical harmonics functions [21]. The B-Format is the first-order Ambisonics, which encodes the directional information of a given three-dimensional sound field into four channels called W, X, Y, Z, where W is the omini-directionial channel, X, Y and Z are the three figure-of-eight directional channels [22]. For a point sound source $p(n)$ assimilated to a plane-wave coming from azimuth $\phi$ and elevation $\delta$, where $n$ denotes the time index, the decomposition of the sound field on these four channels can be formulated as follows [22,23]:

$$
\begin{bmatrix} W(n) \\ X(n) \\ Y(n) \\ Z(n) \end{bmatrix} = \begin{bmatrix} \frac{1}{\sqrt{2}} \\ \cos(\phi)\cos(\delta) \\ \sin(\phi)\cos(\delta) \\ \sin(\delta) \end{bmatrix} p(n).
\tag{1}
$$

Ambisonic-encoded signals carry the spatial information of the entire sound field. To drive the loudspeaker signals with a particular layout, an Ambisonic decoder is needed to transform the B-format signals into D-format (loudspeaker signals). The decoding matrix used in this paper can be found in [24]. To show the characteristics of the B-format signals and their corresponding echo components received by a microphone intuitively, the waveforms and spectrograms of a randomly chosen sample are shown in Figure 1. In the left panel of Figure 1, from top to bottom, Figure 1a–d plots the time-domain W-, X-, Y- and Z-channel signals of B-format recording, respectively, and Figure 1e plots the echo signal. In the right panel of Figure 1, Figure 1f–j plot spectrograms of Figure 1a–e.

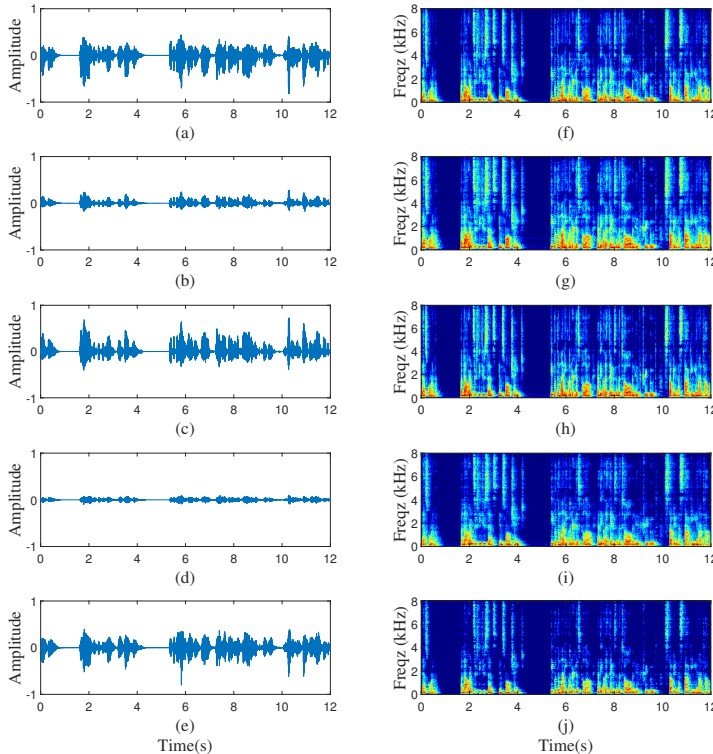

**Figure 1.** Waveforms and Spectrograms of B-format and echo signals. (**a,f**) B-format W-channel signal, (**b,g**) B-format X-channel signal, (**c,h**) B-format Y-channel signal, (**d,i**) B-format Z-channel signal, (**e,j**) echo signal.

## 3. Signal Model

Assuming there are $L$ loudspeakers and one microphone in the near-end room, and there is an Ambisonics recording device in the far-end room, then the typical diagram of the AEC method is shown in Figure 2, and the microphone signal in the near-end room can be formulated as:

$$
\begin{aligned}
y(n) &= \sum_{l=0}^{L} x_l^D(n) * h_l(n) + s(n) + v(n) \\
&= d(n) + s(n) + v(n),
\end{aligned}
\tag{2}
$$

where $x_l^D(n)$ denotes the signals played back by the loudspeakers with $l$ denoting the index of loudspeakers, which can be regarded as the D-format signals in the surround sound system. $h_l(n)$ denotes the room impulse responses (RIRs) from the loudspeakers to the microphone in the near-end room and $*$ denotes the linear convolution operation. The near-end speech signal is denoted as $s(n)$, and $v(n)$ represents the additive noise signal. The echo signals transmitted from the loudspeakers to the microphone are denoted as $d(n)$. In the far-end room, as shown in Figure 2, the Ambisonics recordings are denoted as $x_m^B(n)$, with $m$ denoting the index of Ambisonics input channels, which are generated by the source signal $r(n)$ via the acoustic paths characterized by the impulse responses $g_m(n)$. $x_m^B(n)$ can be decoded to the loudspeaker signals $x_l^D(n)$ via a decoding matrix. The adaptive filtering-based AEC methods attempt to cancel out the echo signal $d(n)$ from $y(n)$ by subtracting the estimated echo signal $\hat{d}(n)$ via the adaptive filters. However, it is difficult for a conventional AEC method to entirely reduce the echo signals and the residual echo signal inevitably exists in the estimated near-end signal $\hat{s}(n)$. Besides, in the surround sound condition, the conventional AEC methods may suffer from the non-unique solution problem due to the high cross-correlation between the loudspeaker signals.

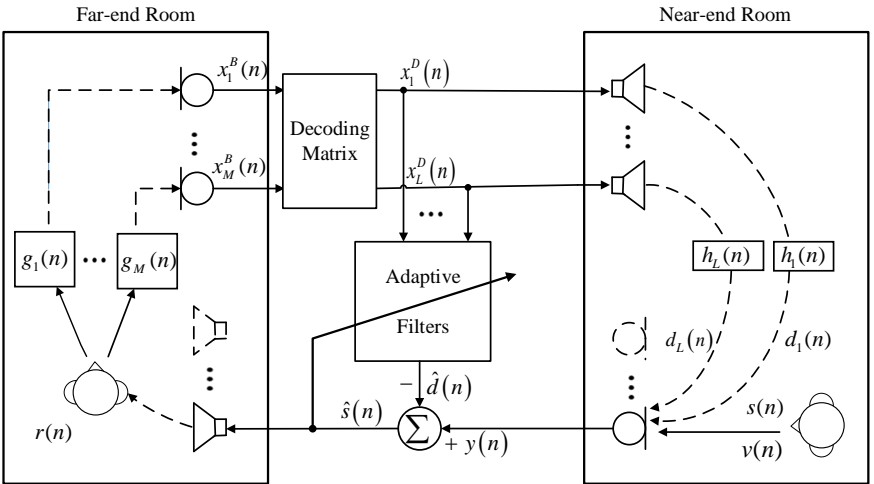

**Figure 2.** Diagram of a surround sound acoustic echo system.

## 4. Proposed GCRN-Based Surround AEC

To recover the near-end signal from microphone recordings directly without decorrelating the far-end signals, a GCRN model-based algorithm was proposed as shown in Figure 3. Note that we chose the B-format signals together with the microphone signals as the inputs of the neural network. In Figure 3, $X_{m,R}^c$ and $X_{m,I}^c$ represent the real and imaginary part of the compressed complex spectrum of B-format signal $x_m^B(n)$, respectively. They can be defined as follows:

$$
X_{m,R}^c = |X_m^B|^\beta \cdot \cos(\theta), \quad X_{m,I}^c = |X_m^B|^\beta \cdot \sin(\theta),
\tag{3}
$$

where $X_m^B$ denotes the complex spectra of $x_m^B(n)$, $|X_m^B|$, and $\theta$ represents the magnitude and phase information of $X_m^B$, respectively. The value of the power compressed coefficient $\beta$ was set as $1/2$ in this work [25,26]. Accordingly, $Y_R^c$, $Y_I^c$ and $\hat{S}_R^c$, $\hat{S}_I^c$ denote the real and imaginary part of the spectra of the microphone and estimated near-end speech signal correspondingly.

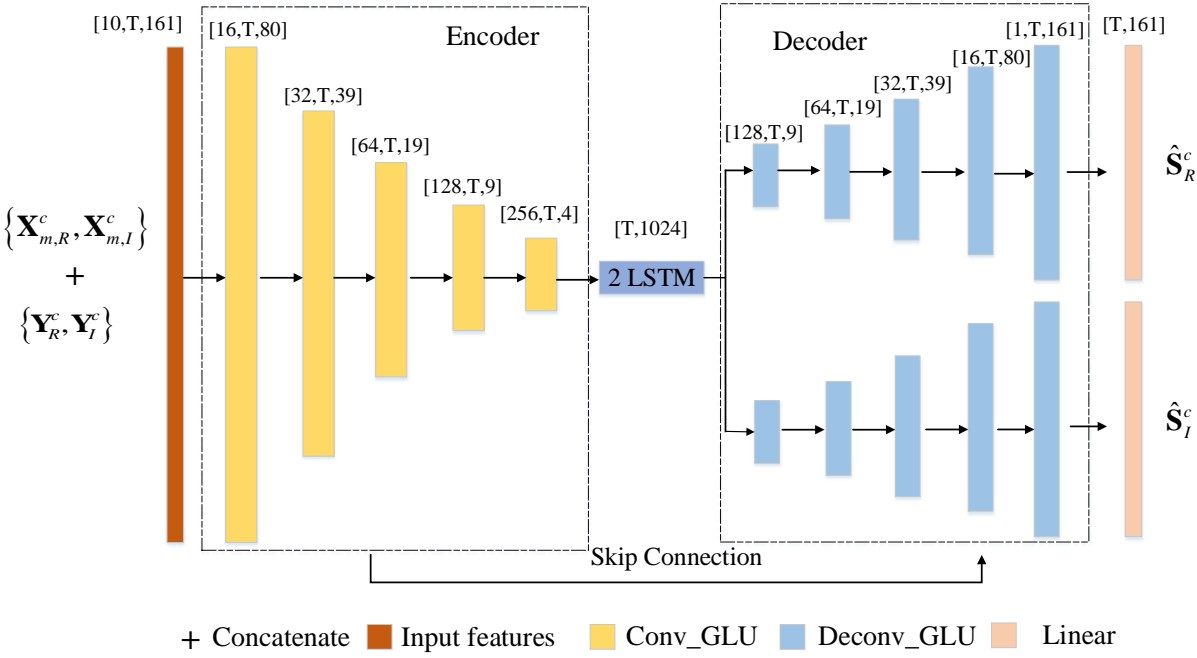

**Figure 3.** The architecture illustration of the GCRN model for surround AEC.

### 4.1. Feature Extraction and Signal Reconstruction

In the GCRN model, the compressed complex spectra of the microphone signal and the four-channel B-format signals were used as the input features. All signals were sampled at 16 kHz. A 20 ms Hanning window with 50% overlap between adjacent frames was used to produce a set of time frames in this work, and a 320-point STFT (short-time Fourier transform) was adopted to generate input features, leading to a 161-dimensional spectral feature in each frame. The compressed complex spectra of the microphone signal $y(n)$ and the four-channel B-format signals $x_m^B(n)$ were concatenated as input feature maps, which have a shape of $[10, T, 161]$, where $T$ represents the total frames. They were taken as the input features to train the GCRN model. As for the output, the real and imaginary parts of the compressed complex spectrum of near-end speech $s(n)$ were used. The loss function can be formulated as follows:

$$\mathcal{J}\left(S^c, \hat{S}^c\right) = 0.5 \cdot \left(|S_R^c| - |\hat{S}_R^c|\right)^2 + 0.5 \cdot \left(|S_I^c| - |\hat{S}_I^c|\right)^2. \tag{4}$$

In the enhancement stage, the estimated real and imaginary parts were used to recover the spectrum near-end signal, which was then used to reconstruct the estimated near-end signal in the time domain by the inverse STFT and the overlap-add method. The estimated magnitude and phase information is given by

$$|\hat{S}^c| = \left(\sqrt{|\hat{S}_R^c|^2 + |\hat{S}_I^c|^2}\right)^{\frac{1}{\beta}}, \hat{\theta} = \arctan\left(\frac{\hat{S}_I^c}{\hat{S}_R^c}\right), \tag{5}$$

finally, the spectrum can be written as:

$$\hat{S}^c = |\hat{S}^c| e^{j\hat{\theta}}, \tag{6}$$

where $|\hat{S}^c|$ and $\hat{\theta}$ are the estimated magnitude and phase information of the near-end speech $s(n)$, respectively.

### 4.2. Model Architecture

In this paper, the GCRN model was used to recover the near-end speech, which is mainly comprised of three components, namely one convolutional encoder, LSTM modules, and two decoders for both real and imaginary part reconstruction. The encoder consists of five convolutional-gated linear units (Conv-GLUs) layers [27], which extract the high-level dimension feature patterns from the inputs, while the decoder serves as the mirror version of the encoder to gradually recover the original size [28]. Each decoder has five deconvolutional-gated linear units (Deconv-GLUs). Each convolution or deconvolution layer is successively followed by a batch normalization (BN) [29] and an exponential linear unit (ELU) [30] activation function. Between the encoder and the decoder, two LSTMs are stacked to effectively establish the sequential correlation between adjacent frames. Additionally, the skip connection [31] was also adopted to concatenate the output of each encoder layer to the input of the corresponding decoder layer, which effectively mitigates the information loss. One linear layer is stacked in the end of each decoder to obtain the real or imaginary estimation.

The detailed parameter setup of the GCRN architecture is presented in Table 1. The $featureMaps \times timeSteps \times frequencyChannels$ format was used to specify the input size and output size of each layer. The (kernelSize, strides, outChannels) format was adopted to represent the hyperparameters of each layer. The number of input feature maps in each decoder layer is doubled because of the skip connections.

**Table 1.** Detailed parameter setup for GCRN.

|  | Layer Name | Input Size | Hyperparameters | Output Size |
|---|---|---|---|---|
| **Encoder** | conv2d_1 | $10 \times T \times 161$ | $1 \times 3, (1,2), 16$ | $16 \times T \times 80$ |
|  | conv2d_2 | $16 \times T \times 80$ | $1 \times 3, (1,2), 32$ | $32 \times T \times 39$ |
|  | conv2d_3 | $32 \times T \times 39$ | $1 \times 3, (1,2), 64$ | $64 \times T \times 19$ |
|  | conv2d_4 | $64 \times T \times 19$ | $1 \times 3, (1,2), 128$ | $128 \times T \times 9$ |
|  | conv2d_5 | $128 \times T \times 9$ | $1 \times 3, (1,2), 256$ | $256 \times T \times 4$ |
|  | reshape_1 | $256 \times T \times 4$ | - | $T \times 1024$ |
|  | LSTM_1 | $T \times 1024$ | 1024 | $T \times 1024$ |
|  | LSTM_2 | $T \times 1024$ | 1024 | $T \times 1024$ |
|  | reshape_2 | $T \times 1024$ | - | $256 \times T \times 4$ |
| **Decoder1(2)** | skip_connection_1 | $256 \times T \times 4$ | - | $512 \times T \times 4$ |
|  | deconv_1 | $512 \times T \times 4$ | $1 \times 3, (1,2), 128$ | $128 \times T \times 9$ |
|  | skip_connection_2 | $128 \times T \times 9$ | - | $256 \times T \times 9$ |
|  | deconv_2 | $256 \times T \times 9$ | $1 \times 3, (1,2), 64$ | $64 \times T \times 19$ |
|  | skip_connection_3 | $64 \times T \times 19$ | - | $128 \times T \times 19$ |
|  | deconv_3 | $128 \times T \times 19$ | $1 \times 3, (1,2), 32$ | $32 \times T \times 39$ |
|  | skip_connection_4 | $32 \times T \times 39$ | - | $64 \times T \times 39$ |
|  | deconv_4 | $64 \times T \times 39$ | $1 \times 3, (1,2), 16$ | $16 \times T \times 80$ |
|  | skip_connection_5 | $16 \times T \times 80$ | - | $32 \times T \times 80$ |
|  | deconv_5 | $32 \times T \times 80$ | $1 \times 3, (1,2), 1$ | $1 \times T \times 161$ |
|  | reshape_3 | $1 \times T \times 161$ | - | $T \times 161$ |
|  | Linear | $T \times 161$ | 161 | $T \times 161$ |

## 5. Experimental Results and Discussions

### 5.1. Experiment Settings

The English reading speech of the DNS-challenge dataset [32] was used as the near-end and the far-end reference signals to perform experiments in the surround AEC situation, which was derived from Librivox, consisting of 65,348 clean clips from 1948 speakers. Each speaker was taken from about 33 utterances with about 30 s for each. Eighty percentages of speakers were used for training and the remaining twenty percentages for testing. For

each training sample, two speakers were randomly selected, and one was used as the far-end speaker and the other was used as the near-end speaker. We randomly cut 12 s long segments from each utterance of the far-end speaker, and the segment was used as the far-end speech. For the near-end speech, we randomly cut 3 s long segments from each utterance of the near-end speaker, and this segment was zero-padded on both sides to ensure that each near-end speech had the same duration as the far-end speech. The zero-padding operation was used to simulate the single-talk scene in real-world scenarios.

The RIRs were generated using the image method [33]. In the far-end room, four microphones with different directivities were used to simulate an Ambisonics recording device located in the center of the room, including an omni-directional microphone, dubbed W, and three figure-of-eight directional microphones, dubbed X, Y and Z, respectively. Signals recorded by these four microphones constitute the B-format signal. Thus, for each sound source and Ambisonics microphone pair, we can get four RIRs, which were then defined as an Ambisonic RIRs group. In the near-end room, the RIRs were generated according to the position of the microphone, its directivity, and loudspeakers. To improve the generalization capacity of the DNN, the length and width of the simulated rooms were randomly sampled from $[3, 10]$ m with 1 m interval and the height was sampled form $[3, 5]$ m. In each simulated room, the reverberation time $RT_{60}$ value was randomly sampled from $\{0.3\,\text{s}, 0.5\,\text{s}, 0.6\,\text{s}, 0.7\,\text{s}, 0.9\,\text{s}\}$. The locations of the microphones and loudspeakers in the simulated room are shown in Figure 4, where the 5.1 surround sound system without the center and subwoofer channels was taken into consideration and the microphones were fixed in the center of each simulated room. As illustrated in Figure 4, four surround loudspeakers $p_1$–$p_4$ were used and $\alpha$ represents the angle between loudspeakers and the horizontal axis. $\lambda$ represents the distance between the loudspeakers and microphones. For standard (ITU-R BS 77) 5.1 surround set-up [34], the $\alpha$ angles for $p_1$–$p_4$ were set to $\{190°, 120°, 60°, 350°\}$. The nonstandard 5.1 surround set-up was also considered. In the nonstandard conditions, the $\alpha$ angle for loudspeaker $p_1$ was randomly sampled from $[190°, 260°]$ with $10°$ interval, and similarly, the $\alpha$ ranges for $p_2$, $p_3$, $p_4$ were set to be $[100°, 170°]$, $[10°, 80°]$ and $[280°, 350°]$ with $10°$ interval, respectively. $\lambda$ was sampled from $\{1\,\text{m}, 1.2\,\text{m}, 1.5\,\text{m}\}$. In the far-end room, the angle between the speaker and horizontal axis was randomly sampled from $[10°, 360°]$ with $10°$ interval, and the distance between the speakers and microphones was sampled from $\{0.3\,\text{m}, 0.5\,\text{m}, 0.7\,\text{m}, 1\,\text{m}, 1.2\,\text{m}\}$. Besides, the height of the microphones, loudspeakers and sound sources were all set to 1.2 m.

The far-end speech was convolved with four RIRs, which are from the same Ambisonic RIRs group as mentioned above, resulting in the four-channel Ambisonic B-format signals. The B-format signals were then decoded to the D-format signals by using the decoding rules as described in [21,24]. The echo signals were generated by convolving the D-format signals with randomly selected RIRs. Finally, the near-end speech was mixed with the echo signals under a signal-to-echo ratio (SER) value randomly selected from $\{0, 5, 10, 15\}$ dB to get the near-end microphone signals. The SER is evaluated on double-talk scenarios, which is defined as:

$$\text{SER} = 10\log_{10}\left(\frac{\sum_n s^2(n)}{\sum_l \sum_n d_l^2(n)}\right). \tag{7}$$

As described in [13], white Gaussian noise was also taken into consideration as the microphone internal noise and the signal-to-noise ratio (SNR) was set as 30 dB, which is defined as:

$$\text{SNR} = 10\log_{10}\left(\frac{\sum_n s^2(n)}{\sum_n v^2(n)}\right), \tag{8}$$

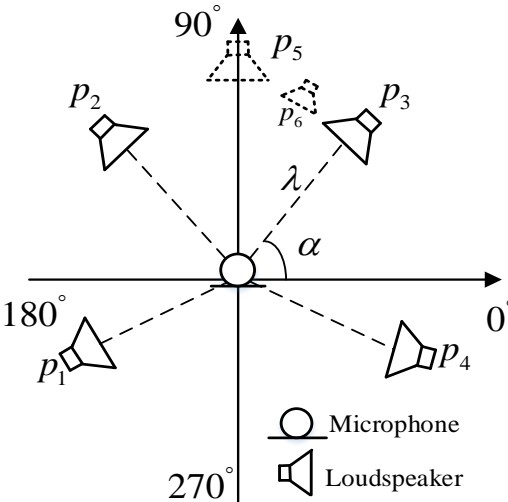

**Figure 4.** Location of microphones and loudspeakers.

The mean squared error (MSE) between the estimated and true compressed complex spectra of the near-end signal was used as a loss function, and optimized by the Adam algorithm [35]. The learning rate was set to $3 \times 10^{-4}$. The mini-batch size was set to 16 at the utterance level  and each batch took about 1.20 s in the training stage. The GCRN network was trained for 50 epochs. All training samples were padded with zeros to have the same number of time frames as the longest sample within a mini-batch and cross-validation was used to select the best model and prevent overfitting.

In single-talk scenarios, the AEC performance is evaluated in terms of ERLE and it is defined as:

$$\text{ERLE} = 10\log_{10}\left(\frac{\sum_n y^2(n)}{\sum_n \hat{s}^2(n)}\right). \tag{9}$$

In double-talk scenarios, PESQ is used to evaluate the AEC performances of different algorithms. PESQ is a widely used speech quality metric which ranges from$-0.5$ to 4.5, and highly correlates with subjective scores [20]. For both the ERLE and PESQ metrics, a higher score indicates better performance.

The least mean square (LMS) algorithm is one of the most widely used adaptation methods in the echo path identification. Among LMS algorithms, the partitioned block frequency domain LMS (PBFDLMS) algorithm is popular for its lower computational complexity than the time-domain LMS algorithms and its lower latency than frequency-domain LMS algorithms, especially when the acoustic echo path is relatively long [14]. To validate the proposed algorithm, the PBFDLMS algorithm with Wiener post-filtering was chosen as the baseline. The PBFDLMS algorithm should be combined with the double-tale detector (DTD) algorithm [36,37]. In order to reduce the performance degradation caused by the DTD method, an ideal DTD was assumed for the PBFDLMS algorithm. The test mixtures were generated in the same way, which were not used in the training procedure.

*5.2. Performance and Analysis*

The performance of the proposed GCRN-based AEC algorithm trained with B-format data (denoted as B-format Model) was firstly analyzed through comparison with the traditional PBFDLMS with the Wiener post-filtering method. Besides, we also trained a model using the D-format signals as the reference signal (denoted as D-format Model). For this model, the standard 5.1 surround sound set-up, as mentioned in Section 5.1, was used, and the decoding rules for this model were fixed. As for the B-format model, the decoding rules are in accordance with the loudspeaker layouts. Moreover, a model using only one

channel B-format data and the microphone signal (denoted as Singlechn Model) as inputs was also performed to compare with the B-format model.

The performance of the B-format and D-format model was firstly tested in standard 5.1 surround sound set-up condition without the center and the subwoofer channels. One test set generated with standard loudspeaker layouts was used, where the $\alpha$ sets were the same as the ones used to train the D-format model. The PESQ and ERLE results in this situation are shown in Table 2. The $RT_{60}$ value was set to 0.5 s in the far-end room. From Table 2, one can see that in the standard loudspeaker assignment condition, the D-format model outperformed the B-format model in all conditions in terms of the ERLE criterion, because the tested surround sound set-up matched the training dataset of the D-format model completely. Both models had similar performances in terms of the PESQ criterion.

To further evaluate the generalization capability of GCRN-based AEC algorithm to unseen RIRs and loudspeaker layouts, different $\alpha$ settings were used to generate the RIRs in the near-end room. As shown in Figure 4, the $\alpha$ angle for loudspeaker $p_1$ was randomly sampled from $[186°, 261°]$ with $15°$ interval, and similarly, the $\alpha$ ranges for $p_2, p_3, p_4$ were set to be $[99°, 174°]$, $[9°, 84°]$ and $[276°, 351°]$ with $15°$ interval, respectively. The room dimensions were the same as the ones used in the training set. Note that the $\alpha$ settings were different from that in the training sets, thus no RIRs were overlapped between the training and test sets. In this experiment, the $RT_{60}$ value was set to 0.5 s in the far-end room, and we tested the performance of each algorithm in different $RT_{60}$ values $(0.3\,\mathrm{s}, 0.6\,\mathrm{s}, 0.9\,\mathrm{s})$ of the near-end room. The PESQ and the ERLE results for different SER conditions are shown in Table 3. One can find that the GCRN-based surround AEC method performed much better than the other algorithms in every test condition. For the D-format model, the decoding rules mismatched with ones used in the training procedure, which is the main reason for its performance degradation. As for the one-channel model, it seemed that one-channel information was insufficient for the model to achieve the best performance. Besides, compared with Tables 2 and 3, one can find that the D-format model performed much worse in the nonstandard surround set-up, while the B-format model had similar performance in each test condition.

To test the performance of the proposed method in the real acoustic environment, real-world experiments were conducted in one meeting room. The $RT_{60}$ values of the far-end room and near-end room were about 0.8 s and 0.25 s, respectively. The sizes of the two meeting rooms were about $6.2 \times 4.6 \times 2.7$ m$^3$ and $4.2 \times 4.1 \times 3.3$ m$^3$, separately. In the far-end room, the distance between the speaker and Ambisonics microphone was about 0.85 m. The recorded B-format signal in the far-end room was then decoded and played by the loudspeakers in the near-end room. Meanwhile, the echo signal was picked up by a microphone. Here, two different $\alpha$ settings, as illustrated in Figure 4, were used to represent the standard and nonstandard loudspeakers assignment situations and the two $\alpha$ sets for loudspeakers $p_1$–$p_4$ are $\{190°, 120°, 60°, 350°\}$ and $\{225°, 135°, 45°, 315°\}$, separately. As the reverberation of near-end speech was not taken into consideration, the near-end signal was directly mixed with the recorded echo signal at SER = 5 dB to generate the near-end microphone signal. The spectrogram of the real recordings with standard or nonstandard loudspeaker layouts processed by different algorithms are plotted in Figures 5 and 6. Figure 5 represents the standard loudspeakers assignment situation; the nonstandard scene is shown in Figure 6. The ERLE and PESQ scores of two models were also presented in the two figures. As shown in these two figures, both the D-format and B-format models performed well in the standard situation. Note that in the real experiment, the deviation of angle and distance between the loudspeakers and microphone inevitably existed. Although the decoding rules were the same as the training stage, the contribution of each loudspeaker signal for the echo signal was different from the standard condition due to the existence of the deviation. This can be the main reason for the performance degradation of the D-format model. Besides, the characteristics of each loudspeaker can also affect the produced echo. In the unknown decoding rule scene, the performance of

the B-format model was much superior than the D-format model, indicating the greater robustness of the B-format model for practical applications.

**Table 2.** Performance comparisons among different algorithms in different SERs and the standard surround set-up conditions.

| | Algorithms | ERLE (dB) | | | | PESQ | | | |
|---|---|---|---|---|---|---|---|---|---|
| **SER(dB)** | **-** | **0** | **5** | **10** | **15** | **0** | **5** | **10** | **15** |
| RT$_{60}$ = 0.3 s | Unprocessed | - | - | - | - | 1.85 | 2.15 | 2.46 | 2.71 |
| | PBFDLMS | 14.54 | 16.13 | 16.00 | 13.89 | 2.46 | 2.71 | 2.85 | 2.93 |
| | Singlechn Model | 57.94 | 58.88 | 56.44 | 51.96 | 2.66 | 3.02 | 3.27 | 3.51 |
| | D-format Model | **62.30** | **60.72** | **58.40** | **53.60** | 2.81 | 3.15 | **3.37** | 3.63 |
| | B-format Model | 61.49 | 59.91 | 57.57 | 52.30 | **2.85** | **3.18** | 3.34 | 3.63 |
| RT$_{60}$ = 0.6 s | Unprocessed | - | - | - | - | 1.79 | 2.18 | 2.39 | 2.71 |
| | PBFDLMS | 13.63 | 12.54 | 13.04 | 14.08 | 2.44 | 2.57 | 2.74 | 2.97 |
| | Singlechn Model | 56.32 | 57.66 | 56.25 | 52.36 | 2.66 | 3.02 | 3.27 | 3.44 |
| | D-format Model | **58.53** | **59.68** | **57.62** | **53.83** | 2.63 | 2.95 | 3.27 | 3.50 |
| | B-format Model | 57.34 | 58.56 | 56.34 | 52.44 | **2.67** | **3.02** | **3.31** | **3.53** |
| RT$_{60}$ = 0.9 s | Unprocessed | - | - | - | - | 1.86 | 2.09 | 2.42 | 2.72 |
| | PBFDLMS | 11.31 | 13.02 | 12.75 | 13.55 | 2.23 | 2.44 | 2.76 | 2.99 |
| | Singlechn Model | 57.26 | 56.18 | 56.53 | 52.14 | 2.46 | 2.82 | 3.18 | 3.46 |
| | D-format Mode | **58.82** | **58.59** | **56.98** | **53.62** | 2.52 | 2.89 | 3.23 | 3.50 |
| | B-format Model | 57.56 | 56.86 | 56.88 | 52.63 | **2.57** | **2.95** | **3.26** | **3.52** |

**Table 3.** As in Table 2 but for the nonstandard surround set-up conditions.

| | Algorithms | ERLE (dB) | | | | PESQ | | | |
|---|---|---|---|---|---|---|---|---|---|
| **SER(dB)** | **-** | **0** | **5** | **10** | **15** | **0** | **5** | **10** | **15** |
| RT$_{60}$ = 0.3 s | Unprocessed | - | - | - | - | 1.85 | 2.12 | 2.46 | 2.70 |
| | PBFDLMS | 13.74 | 13.89 | 15.00 | 14.01 | 2.36 | 2.69 | 2.82 | 3.00 |
| | Singlechn Model | 59.54 | 59.53 | 56.92 | 52.43 | 2.69 | 3.03 | 3.25 | 3.50 |
| | D-format Model | 53.16 | 55.80 | 55.75 | 52.57 | 2.72 | 3.05 | 3.31 | 3.56 |
| | B-format Model | **63.67** | **61.09** | **58.11** | **52.80** | **2.87** | **3.17** | **3.37** | **3.63** |
| RT$_{60}$ = 0.6 s | Unprocessed | - | - | - | - | 1.74 | 2.18 | 2.39 | 2.71 |
| | PBFDLMS | 12.44 | 14.04 | 14.23 | 14.38 | 2.41 | 2.55 | 2.75 | 2.89 |
| | Singlechn Model | 59.47 | 58.62 | 56.40 | 53.34 | 2.58 | 2.92 | 3.23 | 3.40 |
| | D-format Model | 54.83 | 53.77 | 54.75 | 53.61 | 2.56 | 2.92 | 3.24 | **3.51** |
| | B-format Model | **60.32** | **59.49** | **57.97** | **53.72** | **2.72** | **3.05** | **3.32** | 3.50 |
| RT$_{60}$ = 0.9 s | Unprocessed | - | - | - | - | 1.86 | 2.09 | 2.42 | 2.72 |
| | PBFDLMS | 12.15 | 13.10 | 12.41 | 14.10 | 2.38 | 2.44 | 2.80 | 2.96 |
| | Singlechn Model | 57.44 | 58.54 | 57.31 | 52.76 | 2.51 | 2.87 | 3.21 | 3.47 |
| | D-format Model | 52.27 | 57.94 | 57.32 | 53.26 | 2.51 | 2.87 | 3.21 | 3.49 |
| | B-format Model | **59.30** | **59.68** | **57.65** | **53.52** | **2.65** | **2.98** | **3.29** | **3.54** |

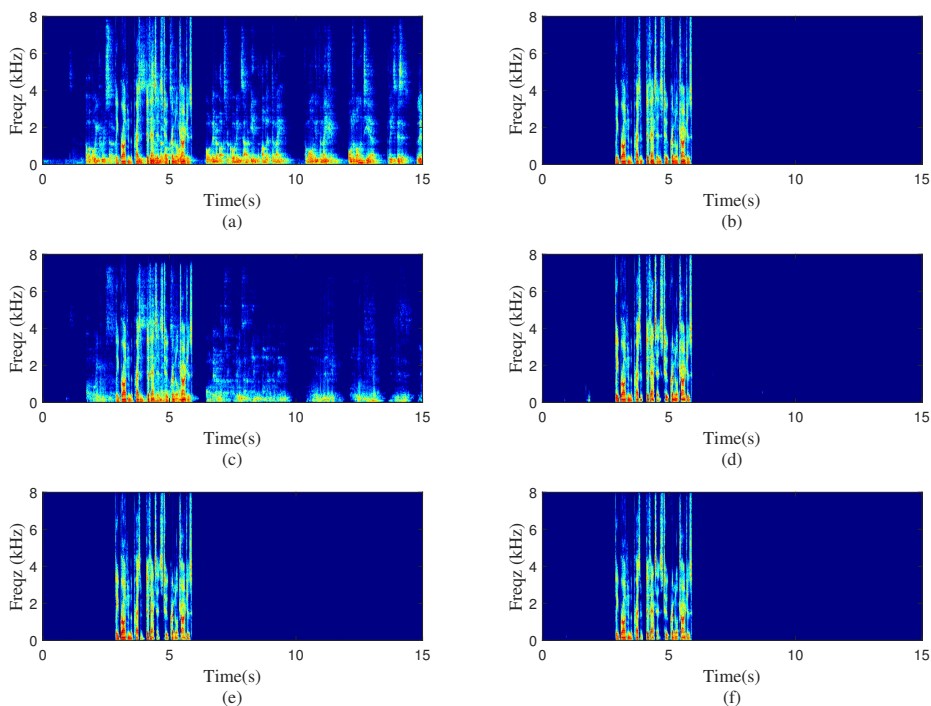

**Figure 5.** Spectrograms processed by different methods with standard loudspeakers layout. (**a**) Microphone signal, PESQ = 2.42, (**b**) clean near-end speech, (**c**) PBFDLMS algorithm, ERLE = 15.04 dB, PESQ = 2.74, (**d**) D-format model-based algorithm, ERLE = 41.92 dB, PESQ = 3.01, (**e**) Singlechn model-based algorithm, ERLE = 56.21 dB, PESQ = 2.84, (**f**) B-format model-based algorithm, ERLE = 58.38 dB, PESQ = 2.98.

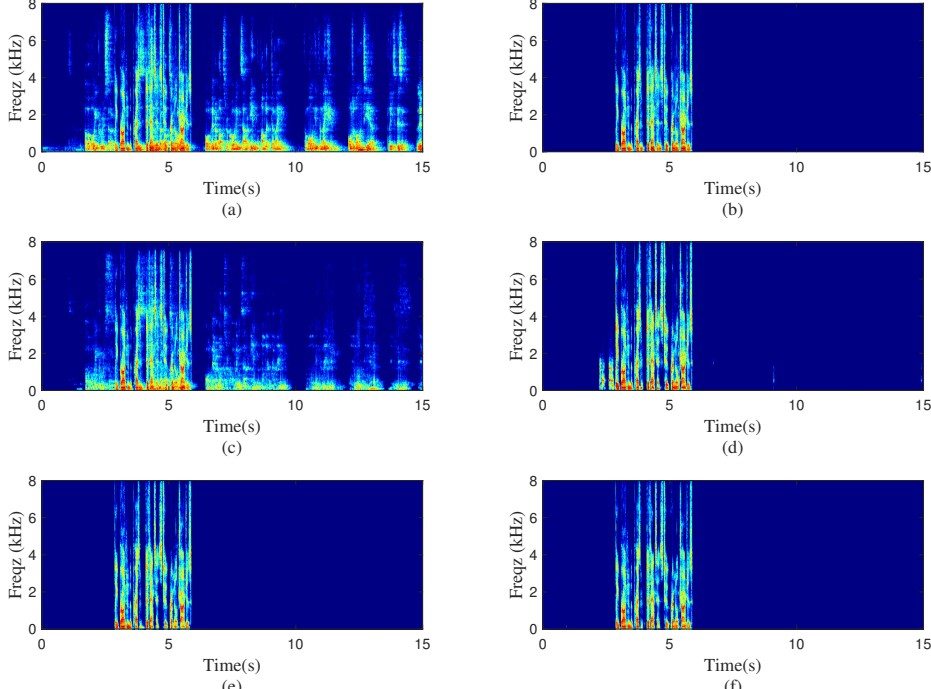

**Figure 6.** Spectrograms processed by different methods with nonstandard loudspeakers layout. (**a**) Microphone signal, PESQ = 2.45, (**b**) clean near-end speech, (**c**) PBFDLMS algorithm, ERLE = 14.07 dB, PESQ = 2.59, (**d**) D-format model-based algorithm, ERLE = 17.71 dB, PESQ = 2.76, (**e**) Singlechn model-based algorithm, ERLE = 52.56 dB, PESQ = 2.85, (**f**) B-format model-based algorithm, ERLE = 54.73 dB, PESQ = 2.97.

## 6. Conclusions

This paper proposed a compressed complex spectrum mapping approach for surround AEC, and the method does not need to identify the acoustic echo paths explicitly, and thus does not suffer from the non-unique solution problem. The proposed method used the B-format signals instead of the far-end D-format loudspeaker signals as the reference signals of the AEC algorithm. Experimental studies showed that the model trained with the B-format signals was more robust than that trained with D-format signals against various loudspeaker layouts, including the standard and nonstandard 5.1 surround set-ups. The proposed algorithm outperformed the traditional PBFDLMS in both the single-talk and double-talk scenarios. Experimental results in real acoustic scenarios further confirmed the effectiveness of this method. In the near future, we will explore more effective network structures, such as Transformer-based networks [38], and compare them with this work.

**Author Contributions:** Conceptualization, G.L. and C.Z.; methodology, G.L. and Y.K.; software and validation, G.L. and Y.K.; writing—original draft preparation, G.L.; writing—review and editing, Y.K. and C.Z.; supervision, X.L.; funding acquisition, Y.K. All authors have read and agreed to the published version of the manuscript.

**Funding:** This research was funded by the National Natural Science Foundation of China under Grant 62101550.

**Institutional Review Board Statement:** Not applicable.

**Informed Consent Statement:** Not applicable.

**Data Availability Statement:** Not applicable.

**Conflicts of Interest:** The authors declare no conflict of interest.

## Abbreviations

The following abbreviations are used in this paper:

| | |
|---|---|
| AEC | Acoustic echo cancellation |
| DNN | Deep neural network |
| BLSTM | Bidirectional long short-term memory |
| CRN | Convolutional recurrent network |
| PBFDLMS | Partitioned block frequency domain least mean square |
| GCRN | Gated convolutional recurrent network |
| PESQ | Perceptual evaluation of speech quality |
| ERLE | Echo return loss enhancement |
| STFT | Short-time Fourier transform |
| Conv-GLUs | Convolutional gated linear units |
| Deconv-GLUs | Deconvolutional gated linear units |
| BN | Batch normalization |
| ELU | Exponential linear unit |
| SER | Signal-to-echo ratio |
| SNR | Signal-to-noise ratio |
| MSE | Mean squared error |
| RIRs | Room impulse responses |

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
