# Peer review of "Deep Learning-Based Acoustic Echo Cancellation for Surround Sound Systems"

_applsci, doi:10.3390/app13031266_

Round 1

Reviewer 1 Report

This paper presented a gated convolutional recurrent network (GCRN) model to suppress the echo signal for the surround sound reproduction system. The B-format signals instead of the far-end D-format loudspeaker signals was used as the reference signals of the AEC algorithm to achieve better generalization capability against different loudspeaker layouts. Experimental results on both the simulated and real acoustic environments showed the effectiveness of the proposed algorithm in surround AEC, and outperformed other competing methods in terms of the speech quality improvement and the echo reduction amount in in both the single-talk and double-talk scenarios. The paper is interesting. Some of the concerns are shown as follows to further refine the paper.

(1)   In line 209, why was the PBFDLMS algorithm chosen as the baseline? Please give a more briefly explanation.

(2)   In line 216, how to generate the training dataset? Please give more details about it.

(3)   In Table 2, in the standard loudspeaker assignment condition, the D-format model outperformed the B-format model in all conditions in terms of the ERLE criterion. However, in real measurement as shown in Fig. 4, the B-format model based algorithm is the best. Please explain the reason. Please also explain why the measurement results (Fig. 5) meet well with the simulated results (Table 3).

Reviewer 2 Report

This paper proposes a deep learning-based AEC method to recover the desired near-end speech from 5 microphone signals in surround sound systems.

The paper should be updated following:

1) The comparison with the state of the state should be improved and better details. A sub-paragraph explaining the differences with the state-of-the-art could help.

2)Is it possible to include a figure explaining the data's nature?

3) I agree with the reviewer about using an encoder/decoder for signal processing, but I wondered if the authors had evaluated other approaches.

4)More details about the implementation of the network and training time

5)How do the authors prevent overfitting issues?

Reviewer 3 Report

Good paper, well written and clear results! 

No comments on content, some minor comments on English language and some minor mistakes. 

Abstract, Line 5: Spell out AEC first time: Acoustic Echo Cancellation (AEC)

Line 5: "near-end speech" or "near-field speech"?

Line 11: delete "the", and spelling mistake in reall. should be

Experimental results on both simulated and real acoustic environments showed the effectiveness of...

Line 19 Change to: the echo signal, due to the coupling between microphones and loudspeakers,

Line 58 change to It is well known that Ambisonic technique

Line 65 proposes

Line 67: are not were 

Line 82: no space after conclusions.

Line 107 denoted

Line 192 and 195: angle (not angel) 

Round 2

Reviewer 2 Report

The authors have successfully addressed al reviewers' points.